# Toward Understanding the Transferability of Adversarial Suffixes in Large Language Models

## Abstract

Discrete optimization-based jailbreaking attacks on large language models aim to generate short, nonsensical suffixes that, when appended onto input prompts, elicit disallowed content. Notably, these suffixes are often *transferable*—succeeding on prompts and models for which they were never optimized. And yet, despite the fact that transferability is surprising and empirically well-established, the field lacks a rigorous analysis of when and why transfer occurs. To fill this gap, we identify three statistical properties that strongly correlate with transfer success across numerous experimental settings: (1) how much a prompt without a suffix activates a model's internal refusal direction, (2) how strongly a suffix induces a push away from this direction, and (3) how large these shifts are in directions orthogonal to refusal. On the other hand, we find that prompt semantic similarity only weakly correlates with transfer success. These findings lead to a more fine-grained understanding of transferability, which we use in interventional experiments to showcase how our statistical analysis can translate into practical improvements in attack success.

## 1 Introduction

Adversarial examples—carefully crafted input perturbations that can make models behave in undesirable ways—remain a fundamental obstacle to achieving robustness across deep learning tasks and data modalities (Goodfellow et al., 2015; Carlini and Wagner, 2017; Madry et al., 2017). A particularly puzzling property of these perturbations is their *transferability*—perturbations optimized for one input or model are often effective on others (Szegedy et al., 2014; Papernot et al., 2016).

Although initially discovered in the context of image classification (see, e.g., Salman et al. (2020); Tramèr et al. (2017)), transferability has resurfaced as a key aspect of *jailbreaking* large language models (LLMs) to elicit harmful responses (Wei et al., 2023). While jailbreaks are typically optimized for a particular model and input prompt, recent empirical findings conclusively show that jailbreaks often transfer between models, despite differing architectures and training data (Chao et al., 2023; Andriushchenko et al., 2024). Of particular note are discrete optimization-based jailbreaking algorithms that generate short, nonsensical suffixes that, when appended onto a prompt requesting harmful content, return a compliant response (Zou et al., 2023; Geisler et al., 2024; Wallace et al., 2021). And while the transferability of suffix-based attacks is empirically well-established, the field lacks a fine-grained understanding of when, why, and to what extent transfer occurs for these attacks.

In this paper, we identify features that are predictive of suffix-based transfer success by conducting a statistical and interventional study of the following questions: (1) Why are some prompts more susceptible to suffix-based attacks than others; (2) Which properties of a given suffix lead to successful transfer; and (3) What internal model mechanisms govern transfer success? Our study of these questions includes analysis of *intra-model transfer*—generalization across prompts within the same model—and *inter-model transfer*—generalization across models with the same prompt. Our main findings, which rely on notions related to *refusal directions* (Arditi et al., 2024), are as follows:

- **Prompt refusal connection:** Prompts corresponding to activations that are less aligned with a model's refusal direction are easier to successfully jailbreak, leading to more transfer.
- **Suffix push and orthogonal shift:** Suffixes that successfully transfer tend to induce both antiparallel and orthogonal shifts away from a model's refusal direction.

- **Prompt semantic similarity.** Prompt semantic similarity only weakly predicts transfer, which suggests that the geometry of suffix activation spaces is only loosely tied to linguistic form.

Based on our large-scale statistical analysis, which involves the optimization of 10,000 adversarial suffixes per model, we find that three mechanistic factors contribute to transfer success: refusal connectivity (Def. 4), suffix push (Def. 5), and orthogonal shift (Def. 6). While variants of these quantities have appeared in prior work, our focus is to rigorously measure their effect on transferability through a statistical and interventional analysis. Moreover, we introduce algorithmic interventions that improve the success rates of existing attacks; we hope that this analysis informs the design of future attacks and defenses.

## 2 RELATED WORK

**Transferability of adversarial examples.** Over the past decade, the transferability of adversarial attacks has been observed across data modalities, architectures, and training schemes (Goodfellow et al., 2015; Neekhara et al., 2019; Carlini and Wagner, 2018; Taori et al., 2019; Ren et al., 2019). This finding has prompted various theories that seek to diagnose when and why transferability succeeds, particularly in the context of computer vision. While Tramèr et al. (2017) identify distributional conditions that lead to transfer in linear and quadratic models, Demontis et al. (2019) contend that other factors, including model complexity and gradient similarity, influence transferability. On the other hand, Ilyas et al. (2019) find that different models tend to learn similar non-robust features, making them susceptible to transfer attacks. In contrast to existing research, we provide a statistical and interventional study, which (a) concerns language, rather than images, and (b) identifies distinct features behind transferability based on a mechanistic interpretability analysis of activation spaces (Arditi et al., 2024).

**Transferability of jailbreaks.** The discovery that many distinct jailbreak strategies induce transfer across LLMs has renewed interest in model security (Jain et al., 2023; Robey et al., 2024; Zou et al., 2024). While these varied attack modalities have helped identify model blind spots, this diversity also complicates the task of identifying the principles underlying the success of transferability. To this end, we focus on *suffix-based* jailbreaks (Liu et al., 2023; Zhu et al., 2023; Jones et al., 2023), since they admit structure that facilitates decoupling the effect of the prompt and the suffix. Because attacks from this family are all structurally similar, in this paper, we focus on the most frequently used, well-studied variant: Greedy Coordinate Gradient (GCG) (Zou et al., 2023).

**Mechanistic analyses of model safety.** Our results focus on a mechanistic analysis of jailbreak transferability, building on previous works that give a mechanistic interpretation of model safety. Most relevant is the work of Arditi et al. (2024), who identify a "refusal vector"—a direction in activation space that, when subtracted, reduces refusal on harmful prompts and, when added, triggers refusal on harmless ones. Follow-up studies further demonstrate that different jailbreak strategies alter the model's internal representation of harmfulness in distinct ways (Ball et al., 2024), often making harmful prompts appear more similar to benign prompts (Jain et al., 2024; Lin et al., 2024). By contrast, in this paper, we offer statistical and interventional analyses of the mechanisms behind transferability, which lead to a finer-grained understanding of when and why transfer succeeds.

## 3 SETTING THE STAGE: DEFINITIONS AND FEATURES

We next define preliminary quantities used throughout the paper, and formally define features of prompts and suffix that we analyze in this paper.

### 3.1 PRELIMINARIES

We consider two forms of transfer. *Intra-model transfer* measures whether an adversarial suffix $s$, optimized for a particular prompt $p$, also succeeds when applied to different prompts $p'$ on the same model. *Inter-model transfer* measures whether an adversarial suffix $s$, optimized for a particular prompt $p$ and model $m$, also succeeds on a different model $m'$—either on the same prompt $p$ or a new prompt $p'$. To measure these properties, we also define the following:

**Definition 1 (Attack success rate (ASR))** *Given a suffix $s$, let $n^s_{jailbroken}$ denote the number of prompts for which appending $s$ results in a jailbroken response, and let $n^s_{total}$ denote the total*

*number of prompts tested with suffix s. We define the attack success rate (ASR) as:* $ASR(s) := \frac{n^s_{jailbroken}}{n^s_{total}}$.

**Definition 2** (**Refusal direction** (Arditi et al., 2024)) *Given a set containing harmful and harmless prompts, let* $\mathbf{a}^{i,\ell}_{harm}$ *and* $\mathbf{a}^{j,\ell}_{harmless}$ *denote residual stream activation vectors for the final token at layer* $\ell$ *for the* $i$-th *harmful prompt and the* $j$-th *harmless prompt, respectively. The* **refusal direction** $\mathbf{v}^l_{refusal}$ *at layer* $\ell$ *is defined as the difference between the average activations among the prompts, namely*

$$\mathbf{v}^\ell_{refusal} = \left(\frac{1}{n}\sum_{i=1}^n \mathbf{a}^{i,\ell}_{harm}\right) - \left(\frac{1}{m}\sum_{j=1}^m \mathbf{a}^{j,\ell}_{harmless}\right).$$

The refusal direction compares the activations of contrastive pairs of harmful and harmless prompts in order to extract a single vector in representation space that captures the model's internal representation of harmfulness. Consistent with Arditi et al. (2024), we extract the refusal direction at the *optimal layer* (see Appendix A for details). Thus, for brevity, we often do not include the layer index.

## 3.2 INTRODUCING THE FEATURES

Our aim is to study features of prompts and suffixes that correlate with successful transfer. Several of the features we consider are related to the geometry of LLM activation spaces via the so-called *refusal direction* (see Definition 2)—a direction in activation space that triggers refusal when added to harmless prompts and suppresses refusal when subtracted from harmful prompts (Arditi et al., 2024). Before formally defining each quantity in §3.3, we first informally define each quantity of interest.

1. **Semantic similarity of prompts** (Definition 3). Does a suffix $s$ optimized for a prompt $p$ transfer more reliably to another prompt $p'$ when their representations are similar?
2. **Refusal connectivity of the prompt** (Definition 4). Are some prompts more aligned with the refusal direction (e.g., prompts related to concepts emphasized in model alignment), and are prompts aligned with the refusal direction less susceptible to transfer?
3. **Suffix push** (Definition 5). Are suffixes that induce a larger shift in the opposite (antiparallel) direction from the model's refusal direction more likely to transfer?
4. **Orthogonal shift of the suffix** (Definition 6). Are suffixes that induce a larger shift orthogonal to the model's refusal direction more likely to transfer?

Following the large body of work evincing the existence of a refusal direction in various models, the latter three definitions correspond to the following intuitive hypotheses: (a) prompts aligned with the refusal direction are less likely to transfer, (b) suffixes that induce an antiparallel shift are more likely to transfer, and (c) prompts that induce an orthogonal shift are more likely to transfer. In §3.3, we formally define these quantities, which will serve as the central objects of study in §5.

## 3.3 FORMAL DEFINITIONS

We next formalize the quantities informally introduced in §3.2. Note that all activations are extracted at the same layer as the refusal direction (see Appendix A for details).

**Definition 3** (**Semantic similarity**) *The semantic similarity* $sim_{pp'}$ *of two prompts* $p$ *and* $p'$ *is defined as the cosine similarity of some chosen embeddings* $E(p)$ *and* $E(p')$, *namely*

$$sim_{pp'} := \frac{\langle E(p), E(p')\rangle}{\|E(p)\| \cdot \|E(p')\|}.$$

We calculate these embeddings in two different ways—with activations from the model itself and with embeddings extracted from the sentence embedding model "all-mpnet-base-v2" (UKPLab, 2025).

**Definition 4** (**Refusal connectivity**) *Let* $\mathbf{a}^{base}_i$ *denote the residual stream activation vector at the end-of-instruction token for the* $i$-th *harmful prompt. Given a refusal direction* $\mathbf{v}_{refusal}$ *(as defined in Arditi et al. (2024)), the* refusal connectivity *is measured via the quantities*

$$s^{base}_i := \langle \mathbf{a}^{base}_i, \mathbf{v}_{refusal}\rangle \qquad and \qquad cos(\mathbf{a}^{base}_i, \mathbf{v}_{refusal}) = \frac{\langle \mathbf{a}^{base}_i, \mathbf{v}_{refusal}\rangle}{\|\mathbf{a}^{base}_i\| \cdot \|\mathbf{v}_{refusal}\|}.$$

**Definition 5 (Suffix push)** *Let $a_{ij}^{suffix}$ denote the activations for the string $\langle p_i, s_j \rangle$, which represents the concatenation of prompt $i$ with suffix $j$. For a prompt-suffix pair $(i,j)$, the* suffix push *quantifies the change in refusal connectivity when adding a suffix to the prompt, namely*

$$\Delta_{ij}^{push} := \langle \mathbf{a}_i^{base}, \mathbf{v}_{refusal} \rangle - \langle \mathbf{a}_{ij}^{suffix}, \mathbf{v}_{refusal} \rangle.$$

**Definition 6 (Orthogonal shift)** *Let the projection of an activation vector $\mathbf{a}$ onto the refusal direction $\mathbf{v}_{refusal}$ be defined as $\mathbf{p}(\mathbf{a}) := \frac{\langle \mathbf{a}, \mathbf{v}_{refusal} \rangle}{\|\mathbf{v}_{refusal}\|^2} \cdot \mathbf{v}_{refusal}$. The* orthogonal shift *for a prompt-suffix pair $(i,j)$ measures the change in activations perpendicular to the refusal direction, namely*

$$\delta_{ij}^{\perp} := \left\| \left( \mathbf{a}_{ij}^{suffix} - \mathbf{p}(\mathbf{a}_{ij}^{suffix}) \right) - \left( \mathbf{a}_i^{base} - \mathbf{p}(\mathbf{a}_i^{base}) \right) \right\|_2.$$

## 4 EXPERIMENTAL SETUP

This section details the selection of models, the dataset of harmful prompts, the procedure for generating adversarial suffixes, and the approach for evaluating their jailbreaking success.

**Models.** We use Qwen-2.5-3B-Instruct (Qwen et al., 2025), Llama-3.2-1B-Instruct (Meta AI, 2024), Vicuna-13B-v1.5 (Chiang et al., 2023), and Llama-2-7B-Chat (Touvron et al., 2023). While these models are all safety-trained, this list includes models considered easy to jailbreak (e.g., Vicuna) and harder to jailbreak (e.g., Llama-2). This diversity is crucial for assessing the generalizability of our findings across models with different architectures and safety alignment characteristics. A table highlighting relevant aspects of these models is included in Appendix B.

**Data.** We use the JailbreakBench dataset (Chao et al., 2024), which contains 100 harmful questions and answer targets on topics spanning various risk categories as defined by OpenAI's usage policies.

**Generation of adversarial suffixes.** We generate suffixes for each JailbreakBench prompt (Chao et al., 2024) using the GCG algorithm (Zou et al., 2023). For smaller models (Qwen2.5 and Llama 3.2 ), to obtain stable measurements of the statistical quantities outlined above, we generate 100 distinct suffixes per prompt (i.e., 10,000 suffixes per model) by varying GCG's random seed. Due to computational constraints, for larger models (Vicuna and Llama 2), we use a single suffix per prompt, sourced from the JailbreakBench prompt archive (Chao et al., 2023).

**Evaluating jailbreak success.** To evaluate whether jailbreaks succeed, we use a Llama-3-Instruct-70B judge with the system prompt from JailbreakBench following the recommendation of (Chao et al., 2024, Table 1), who evaluated the effectiveness of six commonly used jailbreaking judges.

## 5 ANALYSIS OF THE FACTORS CORRELATED WITH TRANSFER

Toward understanding the effect of each quantity introduced in §3.2, we first record basic transfer statistics (§5.1). We next qualitatively and quantitatively analyze each quantity (§5.2, §5.3, and §5.4). We then provide a joint statistical analysis to estimate the *predictive strength* of the factors (§5.5). We conclude with an exploration of how these insights can be used to produce more transferable suffixes.

### 5.1 QUALITATIVE ANALYSIS OF TRANSFER STATISTICS

As a preliminary step, we highlight some illustrative properties of transfer that can be gleaned from the raw statistics of suffix-based transfer. We consider three scenarios: intra-model transfer (Figure 1), inter-model transfer (Figure 3), and the impact of multiple random initializations (seeds) on suffix generation and jailbreak success (Figure 2).

**Model susceptibility to jailbreaking.** Figure 1 reveals that models exhibit different susceptibilities to adversarial suffixes. Specifically, Vicuna and Qwen show substantially higher success rates for intra-model suffix transfer compared to the Llama models—suggesting varying levels of inherent vulnerability among these model. The use of multiple seeds for suffix generation (Figure 2) offers a more robust assessment of model jailbreakability. By generating 100 suffixes per prompt using different random initializations, we observe that Qwen consistently shows a higher density of

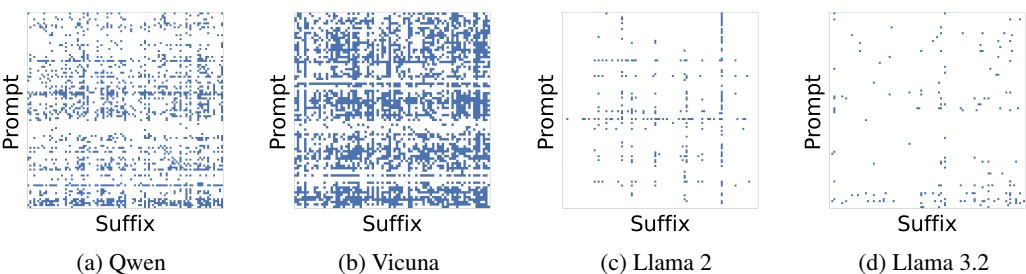

Figure 1: Intra-model transfer with one suffix per prompt for different models. Cells are colored when a suffix successfully jailbreaks a prompt.

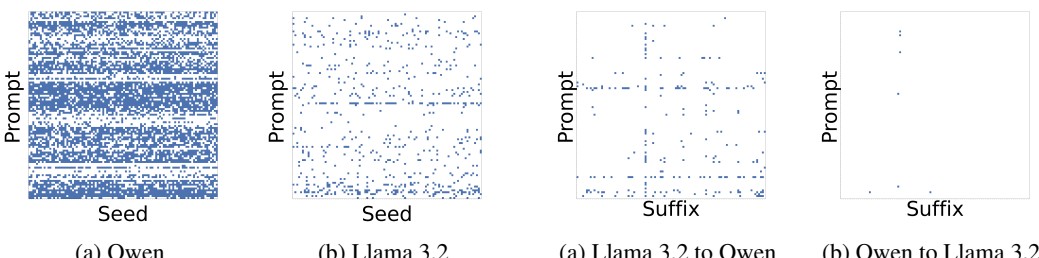

Figure 2: Intra-model transfer with multiple suffixes per prompt. Cells are colored when the corresponding suffix of the seed jailbreaks a prompt.

Figure 3: Inter-model transfer between Llama 3.2 and Qwen. Cells are colored when a suffix successfully jailbreaks a prompt.

successful jailbreaks compared to Llama 3.2. This approach mitigates the effect of single, potentially unrepresentative suffix generations and provides a more stable comparison of model vulnerability.

**Intra-model transferability.** Within individual models, the success of adversarial suffixes is not uniform. Both Figure 1 and Figure 2 highlight that certain prompts are consistently more vulnerable; these appear as horizontal bands with a higher density of successful jailbreaks in the figures. Conversely, some adversarial suffixes exhibit greater potency, successfully compromising a larger set of prompts within the same model. These are identifiable as denser vertical bands in Figure 1. A noteworthy phenomenon is the off-target efficacy of some suffixes: a suffix optimized for a specific prompt (i.e. its corresponding diagonal entry in Figure 1) may fail to jailbreak its prompt but successfully jailbreak other prompts (off-diagonal) within the same model.

**Inter-model transferability.** Suffixes also transfer across models (Figure 3). Using suffixes sampled from the multi-seed pool (Figure 2), we observe an asymmetry: suffixes optimized on a more aligned model (Llama 3.2) transfer better to a less aligned one (Qwen) than vice versa.

**Takeaways.** In sum, transfer occurs within and across models, but success depends on the model, the prompt's vulnerability, and the potency of the suffix. The next sections analyze these factors.

### 5.2 SEMANTIC SIMILARITY

As outlined in §3.2, we aim to determine whether the semantic similarity between the embeddings of two prompts $p$ and $p'$ is predictive of the transferability of a suffix originally optimized for $p$.

**Statistical analysis setup.** We set up a quantitative framework for estimating the effect of semantic similarity ($\text{sim}_{pp'}$, Definition 3) on transferability. For models with multiple suffixes per prompt (Qwen, Llama 3.2), we fit a linear regression model, which predicts the fraction of the average transfer success of a prompt pair $(p, p')$. Hence, we predict $y_{pp'} \in [0, 1]$ from the feature vector $\mathbf{x}_{pp'} := [1, \cos(E(p), E(p'))]$, where $y_{pp'} = 1$ if all suffixes optimized for $p$ jailbreak $p'$ and vice versa. The features are standardized to have mean 0 and variance 1. For models for which we have a single suffix per prompt (Vicuna, Llama 2), we fit an ordinal logistic regression model on the same

Table 1: Regression coefficients (standardized) predicting transfer success based on semantic similarity of prompt embeddings.

| Model | Embedding | $N_{\text{suffix}}$ per prompt | Std. Coef (Odds Ratio) | N | Statistical model |
|---|---|---|---|---|---|
| **Qwen** | Model | 100 | 0.10*** | 1.000.000 | linear reg. |
| | Indep. | 100 | 0.25*** | 1.000.000 | linear reg. |
| **Llama 3.2** | Model | 100 | 0.23*** | 1.000.000 | linear reg. |
| | Indep. | 100 | 0.09*** | 1.000.000 | linear reg. |
| **Vicuna** | Model | 1 | 0.34*** (1.41) | 100.000 | ordinal log. reg. |
| | Indep. | 1 | 0.42*** (1.53) | 100.000 | ordinal log. reg. |
| **Llama 2** | Model | 1 | -0.15*** (0.86) | 100.000 | ordinal log. reg. |
| | Indep. | 1 | 0.18*** (1.2) | 100.000 | ordinal log. reg. |

*Note:* Coefficients are standardized; OR = Odds Ratio; Stars denote statistical significance levels. $^{*}$ $p < 0.05$, $^{**}$ $p < 0.01$, $^{***}$ $p < 0.001$.

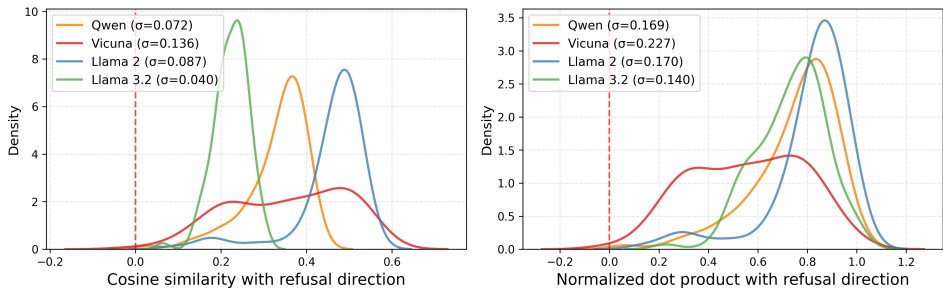

Figure 4: Distribution spread comparison of cosine similarity and (normalized) dot product activations with refusal direction across models.

feature vector but on $y_{pp'} \in \{0, 0.5, 1\}$—0 if neither of the suffixes transfers, 0.5 if only one does, and 1 if both do. Table 1 shows the resulting regression coefficients; following the standard statistical rules-of-thumb (Cohen, 2013; Chen et al., 2010), we conclude that the effect sizes are small.

### 5.3 QUALITATIVE ANALYSIS OF INDIVIDUAL FEATURE EFFECTS

We next *qualitatively* identify key geometric features that are correlated with jailbreak success, deferring a *quantitative* statistical analysis of these features until §5.4.

**Refusal connectivity.** In Figure 4, we plot the density of the cosine similarities and (normalized) dot products over the prompts with the refusal direction for the models we are considering. Vicuna, the most jailbreakable model, has the largest spread, which could explain why the model is not capable of refusing some of the harmful questions without appending a suffix. The distributions are more concentrated for the other models, but there is still a reasonable spread in terms of the component along the refusal direction. In the statistical analysis, we will see how this variance in refusal connectivity is related to whether a suffix jailbreaks a prompt or not.

**Suffix push.** In Figure 5, we plot the distribution spread of semantic similarity and refusal direction alignment for each model. This reveals sevearal clear patterns. First, the average harmful prompt activation has the highest cosine similarity with the refusal direction (blue line). Furthermore, adding the three *least* successful suffixes (orange lines) only marginally reduces this cosine similarity, while adding the three *most* successful suffixes (green lines) significantly suppresses similarity with refusal.

**Orthogonal shift.** Figure 6 shows a positive relationship between suffix transferability (measured as the ASR over all tested prompts per suffix, see Definition 1) and both the orthogonal shift (Definition 6) and the suffix push (Definition 5). This indicates that the likelihood of a successful transfer increases the more a suffix pushes away from refusal and also if it changes activations orthogonal to refusal. Similar patterns can be observed for the other models in Appendix C.

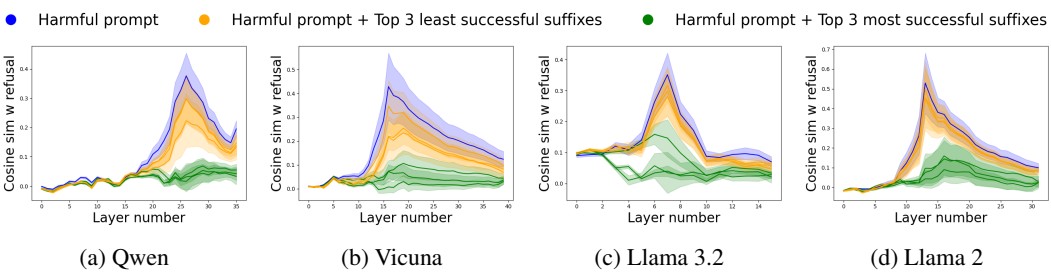

Figure 5: Cross-layer suppression of refusal direction by most and least powerful suffixes for different models, figure based on Arditi et al. (2024). Activations are taken at the end-of-instruction token.

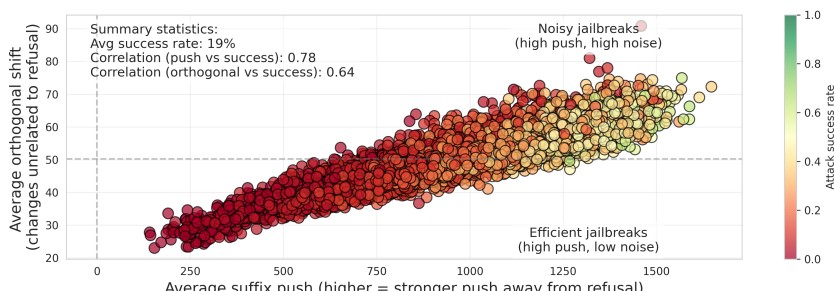

Figure 6: Qwen: Suffix effects on model representations (averaged across prompts for each suffix).

Table 2: Logistic regression coefficients (standardized) predicting transfer success.

| Variable | Qwen | Vicuna | Llama 2 | Llama 3.2 |
|---|---|---|---|---|
| Refusal connec. | $-0.12^{***}$ | $-0.28^{***}$ | $0.21^{**}$ | $-0.06^{***}$ |
| Suffix push | $1.21^{***}$ | $-0.05^{*}$ | $1.53^{***}$ | $0.93^{***}$ |
| Orthogonal shift | $0.97^{***}$ | $0.29^{***}$ | $2.00^{***}$ | $0.82^{***}$ |
| $N$ | $800{,}000$ | $8{,}000$ | $8{,}000$ | $800{,}000$ |

*Note:* Stars denote statistical significance levels. $^{*}$ $p < 0.05$, $^{**}$ $p < 0.01$, $^{***}$ $p < 0.001$.

## 5.4 QUANTITATIVE ANALYSIS OF FEATURE EFFECTS

To quantitatively assess the impact of specific geometric features (defined in §3.2) on transfer, we formulate a logistic regression problem where, for each prompt-suffix pair $(i, j)$, we predict whether the suffix jailbreaks the prompt *solely* from the features of interest. This differs from the semantic similarity setup in §5.2, in that the covariates are prompt-suffix pairs, not prompt-prompt pairs.

**Statistical analysis setup.** For each prompt-suffix pair $(i, j)$, we define a binary target variable $y_{ij} \in \{0, 1\}$, where $y_{ij} = 1$ if suffix $j$ jailbreaks prompt $i$, and $y_{ij} = 0$ otherwise. We consider a logistic regression problem where the covariates are of the form $\mathbf{x}_{ij} := [1, v]$, where $v \in \{s_i^{\text{base}}, \Delta_{ij}^{\text{push}}, \delta_{ij}^{\perp}\}$. Here $s_i^{\text{base}}$ is the refusal connectivity (Def. 4), $\Delta_{ij}^{\text{push}}$ is the suffix push (Def. 5), and $\delta_{ij}^{\perp}$ is the orthogonal shift (Def. 6). The features are standardized to have mean 0 and variance 1. The resulting coefficients indicate the direction of the individual effects on transfer success.

**Results.** The results of the statistical analysis are presented in Table 2. Refusal connectivity has a negative and highly significant effect across all models except Llama 2 (where there is a less statistically-significant positive effect). Hence, refusal connectivity tends to dampen the likelihood of a successful suffix transfer to the prompt. Greater suffix push is associated with higher probability of transfer success for all models but Vicuna (low statistical significance). Lastly, greater orthogonal shift is associated with higher probability of transfer success for all models.

Table 3: Logistic regression coefficients (standardized) predicting transfer success. Darker cell colors indicate larger effect sizes.

| Variable | Qwen | Vicuna | Llama 2 | Llama 3.2 | Llama 3.2 → Qwen | Qwen → Llama 3.2 |
|---|---|---|---|---|---|---|
| Refusal connec. | $-1.43^{***}$ | $-1.37^{***}$ | $-0.22$ | $-0.30^{***}$ | $-1.43^{***}$ | $-0.12$ |
| Suffix push | $2.46^{***}$ | $1.12^{***}$ | $1.34^{***}$ | $0.88^{***}$ | $1.12^{***}$ | $-0.12$ |
| Orthogonal shift | $0.17^{***}$ | $0.27^{***}$ | $1.20^{***}$ | $0.46^{***}$ | $0.93^{***}$ | $0.63$ |
| Interaction effects | | | | ✓ | | |
| Constant | | | | ✓ | | |
| $N$ | 800,000 | 8,000 | 8,000 | 800,000 | 8,000 | 8,000 |
| Pseudo $R^2$ | 0.28 | 0.069 | 0.21 | 0.13 | 0.27 | 0.16 |

*Note:* Stars denote statistical significance levels. $^{*}\ p < 0.05$, $^{**}\ p < 0.01$, $^{***}\ p < 0.001$. Interaction effects include all pairwise interactions between main effects.

## 5.5 ANALYSIS OF JOINT EFFECTS FOR EXPLAINING ADVERSARIAL TRANSFER SUCCESS

In the previous section, we studied the effects of the individual factors of interest correlate with transfer. In this section, we combine them all in a *joint* statistical analysis aimed at determining how different features of the prompt and the suffix affect the likelihood that the suffix successfully jailbreaks the prompt. The joint analysis will allow us to probe the explanatory power of *all* features jointly, their relative effect magnitudes as well as the interdependencies between the features. The analyses focus on all features except semantic similarity given its different covariate setup. However a repetition of the analyses including a related similarity-based feature is in Appendix C.

**Statistical analysis setup.** For each prompt-suffix pair $(i, j)$, we define a binary outcome variable $y_{ij} \in \{0, 1\}$, where $y_{ij} = 1$ if suffix $j$ successfully jailbreaks prompt $i$, and $y_{ij} = 0$ otherwise. To explain $y_{ij}$, we construct a feature vector $\mathbf{x}_{ij}$ capturing the properties of the prompt and the suffix we are interested in (defined in §3.3). Specifically, the feature vector $\mathbf{x}_{ij}$ is given by

$$\mathbf{x}_{ij} := [1, s_i^{\text{base}}, \Delta_{ij}^{\text{push}}, \delta_{ij}^{\perp}, s_i^{\text{base}} \cdot \Delta_{ij}^{\text{push}}, s_i^{\text{base}} \cdot \delta_{ij}^{\perp}, \Delta_{ij}^{\text{push}} \cdot \delta_{ij}^{\perp}]^{\top},$$

where $s_i^{\text{base}} \in \mathbb{R}$ is the refusal connectivity of the prompt (Definition 4), $\Delta_{ij}^{\text{push}} \in \mathbb{R}$ is the suffix push away from refusal (Defition 5), and $\delta_{ij}^{\perp} \in \mathbb{R}$ is the shift orthogonal to refusal (Definition 6). We standardize the coordinates of the feature vector so that they have mean 0 and variance 1. Note that the feature vector includes the individual factors as well as the pairwise products of these terms—this is because we will track the *main effects* due to these factors (i.e. the strength of the dependence of the $\{y_{ij}\}$ on these factors), as well as the *interaction effects* due to pairwise interactions between these factors (i.e. the strength of the pairwise dependence between these factors). This follows classical methodology in statistics (Hastie et al., 2009; Stock and Watson, 2015), according to which the coefficients we fit corresponding to the pairwise interaction effects capture how the influence of one variable changes depending on the value of another variable. This approach hence accounts for non-linear interactions between the main effects. We fit a parameter vector $\boldsymbol{\beta} \in \mathbb{R}^6$ via logistic regression for this setup (i.e. we maximize the likelihood of the labels $\{y_{ij}\}$, such that for a choice of parameters $\boldsymbol{\beta}$, $\mathbb{P}(Y_{ij} = 1)$ is parametrized as $\exp(\boldsymbol{\beta}^T \mathbf{x}_{ij})/[1 + \exp(\boldsymbol{\beta}^T \mathbf{x}_{ij})]$).

**Intra-model transfer results.** The main effects are in line with the single-factor results in §5.3 and §5.4. Higher refusal connectivity is associated with a decreased probability of transfer success; the effect is statistically significant for Qwen, Vicuna and Llama 3.2. Greater suffix push and orthogonal shift are associated with higher probability of transfer success; the effect is statistically significant for all models. Suffix push exhibits the largest effect, followed by refusal connectivity for Qwen and Vicuna, while for the Llama models, refusal connectivity plays a less important role compared to the orthogonal shift. Note that all models include all pairwise interaction effects and a constant. Given that all interaction effects are relatively small compared to the main effects (all below 0.6), we focus on interpreting the main effects. Detailed results are shown in Appendix C.

**Inter-model transfer results.** The logistic regression for inter-model transfer (last two columns in Table 3) shows for Llama 3.2 to Qwen, that the main effects largely mirror the patterns observed in Qwen's intra-model analysis. For Qwen to Llama 3.2 no statistically significant effects were found.

This is likely attributable to the overall very low success rate of transfers in this direction (as seen in Figure 3b), providing insufficient variance for the model to capture significant relationships.

**Takeaways.** In sum, these regression results point to broadly shared mechanisms influencing transfer success, with the suffix push being the most influential factor relative to other predictors (Table 3).

## 5.6 INTERVENTIONAL ANALYSIS

This section shows how our statistical insights can be used as interventions to improve attack success.

**Prompt rephrasing.** Our statistical analysis indicates that prompts more aligned with the refusal direction are harder to jailbreak, reducing suffix transfer. This suggests the following *interventional* experiment: testing whether rephrasing a prompt to be more or less aligned with refusal affects transfer. Using Vicuna, we generate 10 rephrases per prompt, compute their dot product with the refusal direction, and measure how dot product changes relate to ASR changes (see Appendix C for details). We expect a negative relationship as higher dot products should make it harder to transfer, lowering the ASR. Experiments with Qwen and Llama 3.2 confirm this for Qwen (correlation coefficient: -0.08, $p < 0.05$), but not for Llama 3.2 (correlation coefficient: 0.04, $p > 0.05$) due to low statistical significance.

The significant relationship for Qwen suggests that our statistical insights can successfully guide intervention design. For Llama 3.2, while the results were not statistically significant, we note that our rephrasing procedure produced only modest changes in dot products with the refusal direction. We believe more targeted prompt engineering—designed to optimize changes in refusal connectivity—could yield significant results across models, representing a promising direction for future work.

**Altered GCG Loss.** Our statistical analysis indicates that suffixes inducing a larger *suffix push* or *orthogonal shift* are more likely to transfer. This suggests the following *interventional* experiment: modifying the GCG loss to include regularizers favoring suffixes pushing away from or orthogonal to refusal. For these two settings, we evaluate Llama 3.2 with six non-zero regularization coefficients. We use 20 prompts—2 randomly taken from each of the 10 JailbreakBench categories—none of which jailbreak the model without a suffix. For the suffix push regularization term, we generate 100 suffixes for each of the 20 prompts, leading to 40,000 prompt/suffix pairs per coefficient. For the orthogonal shift regularization term, due to computational constraints, we generate 5 suffixes per prompt, leading to 2,000 prompt/suffix pairs per coefficient. We evaluate the ASR of the altered GCG algorithm using our jailbreak judge. We find that for both the suffix push and orthogonal shift regularization terms, the best coefficient is non-zero, corroborating our statistical analyses. Results are presented in Tables 4 and 5.

Table 4: Results for altered GCG loss (Llama 3.2 model): Suffix Push.

| Coefficient | ASR | # jailbroken |
|---|---|---|
| 0 | 0.0138 | 552 |
| 0.00001 | 0.0177 | 709 |
| 0.0001 | 0.0189 | 757 |
| 0.001 | 0.0214 | 855 |
| 0.01 | 0.0176 | 706 |
| 0.1 | 0.0093 | 373 |
| 0.5 | 0.0101 | 406 |

Table 5: Results for altered GCG loss (Llama 3.2 model): Orthogonal Shift.

| Coefficient | ASR | # jailbroken |
|---|---|---|
| 0 | 0.0145 | 29 |
| 0.00001 | 0.0265 | 53 |
| 0.0001 | 0.0195 | 39 |
| 0.001 | 0.0175 | 35 |
| 0.01 | 0.0115 | 23 |
| 0.1 | 0.0020 | 4 |
| 0.5 | 0.0010 | 2 |

## 6 CONCLUSION

Our work identifies prompt- and suffix-specific factors that correlate strongly with successful suffix-based transfer. Through fine-grained statistical analysis, we characterize both the direction and strength of these effects, as well as their interplay. Among suffix-centric factors, the suffix push—the amount of shift away from the refusal direction—plays the strongest role across models. Among prompt-centric factors, the refusal connection—the alignment of a prompt embedding with the refusal direction—plays a strong role for certain models. Together, these factors contribute to a broader conceptual picture linking activations to the mechanisms underlying suffix-based transfer. Finally, through interventional experiments, we also demonstrate that these insights can be used to design stronger attacks and hope they can be used for developing stronger defenses.

## ETHICS STATEMENT

Our work contributes to a fundamental understanding of the vulnerabilites of LLM. While we also show ways of making attacks more successful, we are convinced that our work will contribute to developing technology that is safer to deploy and more aligned with societal benefits.

## REPRODUCIBILITY STATEMENT

We provide extensive implementation details for all experiments—including which models, judges, and datasets we use. We additionally provide the codebase in the supplementary materials file of this submission.

## LLM USAGE STATEMENT

We used Claude Sonnet 4 (Anthropic, 2025) and GPT 4 (OpenAI, 2024) for editing the text and coding assistance.

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

# A    RELEGATED DEFINITIONS FROM SECTION 3.3

**Definition 7 (Optimal layer selection)** *Let $l^* \in \{1, 2, \ldots, L\}$ denote the optimal layer for extracting the refusal direction, where $L$ is the total number of layers in the model. The* optimal layer $l^*$ *is selected as:*

$$l^* = \underset{l \in \{1,2,\ldots,L\}}{\arg\max} \; \textit{Effectiveness}(\mathbf{v}_{refusal}^l) \tag{1}$$

*where Effectiveness$(\mathbf{v}_{refusal}^l)$ measures the success of the refusal direction at layer $l$ in changing model behavior, following* Arditi et al. (2024).

For brevity, in the paper we drop the layer superscript $l^*$ when clear from context. All activations and refusal directions $\mathbf{v}_{\text{refusal}}$, $\mathbf{a}_i^{\text{base}}$, and $\mathbf{a}_{ij}^{\text{suffix}}$ are computed at the optimal layer $l^*$ unless explicitly stated otherwise.

# B    RELEGATED DETAILS FOR MODELS IN PAPER FROM SECTION 4

Table 6: Comparison of model selection

| Attribute | Qwen 2.5 | Llama 3.2 | Vicuna 1.5 | Llama 2 Chat |
|---|---|---|---|---|
| Alignment training | SFT, DPO, GRPO | SFT, DPO, RLHF | SFT | SFT, RLHF |
| Model size | 3B | 1B | 14B | 7B |
| # of generated suffixes | 10.000 | 10.000 | 100 | 100 |

# C    ADDITIONAL QUALITATIVE AND QUANTITATIVE RESULTS RELEGATED FROM SECTION 5

**Additional qualitative results for orthogonal shift**    The following figures show the positive relationship between suffix transferability and both the orthogonal shift and suffix push features for Vicuna (Figure 7), Llama 2 (Figure 8), and Llama 3.2 (Figure 9). The main text includes a similar figure for Qwen (see Figure 6). For all models we observe a similar trend of higher suffix push and higher orthogonal shift being correlated with suffix transferability, albeit with less strong signal for the Llama models. This is because there are less examples of successful transfers in general. The figures for Vicuna and Llama 2 are less dense, given that there is only one suffix per prompt.

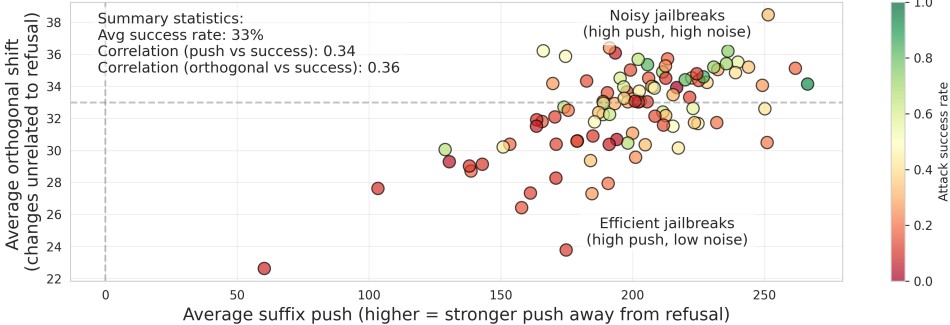

Figure 7: Suffix orthogonal shift and push effects on model representations (averaged across harmful prompts for each suffix ID) for Vicuna.

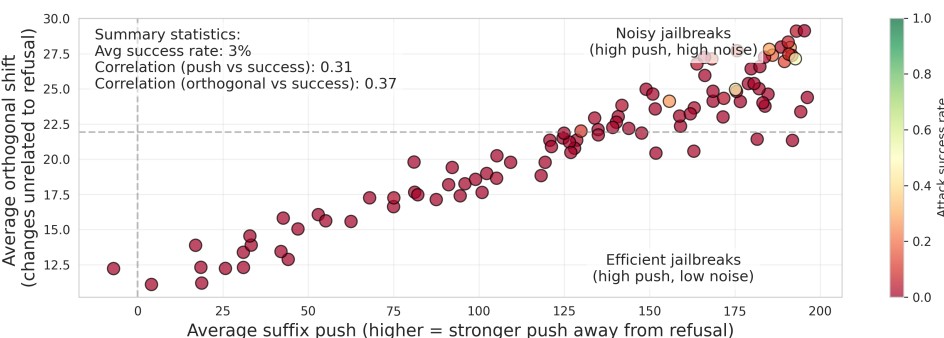

Figure 8: Suffix orthogonal shift and push effects on model representations (averaged across harmful prompts for each suffix ID) for Llama 2.

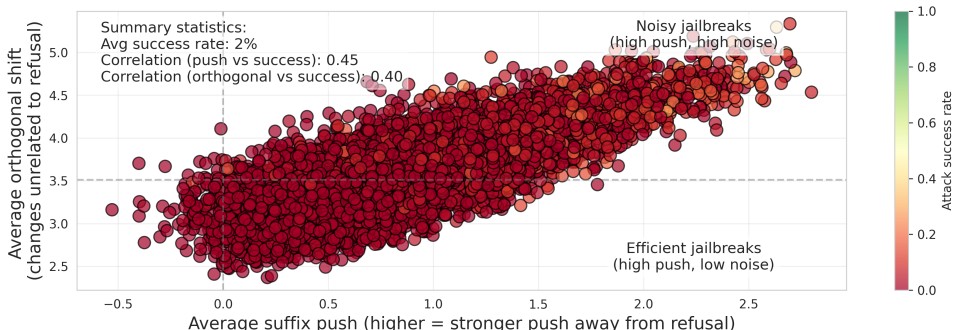

Figure 9: Suffix orthogonal shift and push effects on model representations (averaged across harmful prompts for each suffix ID) for Llama 3.2.

**Additional results for the analysis of joint effects in Section 5.5**  In Section 5.5, we calculate the joint effect of our features of interest in a logistic regression analysis. While Table 3 in the main text focuses on the main effects, Table 7 details the regression coefficients for all interaction effects and the constant.

Table 7: Detailed (standardized) logistic regression coefficients with interaction effects predicting transfer success.

| Variable | Qwen | Vicuna | Llama 2 | Llama 3.2 | Llama 3.2 → Qwen | Qwen → Llama 3.2 |
|---|---|---|---|---|---|---|
| Refusal connec. | $-1.43^{***}$ | $-1.37^{***}$ | $-0.22$ | $-0.30^{***}$ | $-1.43^{***}$ | $-0.12$ |
| Suffix push | $2.46^{***}$ | $1.12^{***}$ | $1.34^{***}$ | $0.88^{***}$ | $1.12^{***}$ | $-0.12$ |
| Orthogonal shift | $0.17^{***}$ | $0.27^{***}$ | $1.20^{***}$ | $0.46^{***}$ | $0.93^{***}$ | $0.63$ |
| Refusal connec. × Suffix push | $0.16^{***}$ | $0.04$ | $-0.46^{***}$ | $-0.30^{***}$ | $0.06$ | $-0.59$ |
| Refusal connec. × Orthogonal shift | $0.47^{***}$ | $0.53^{***}$ | $-0.59^{*}$ | $0.17^{***}$ | $0.35^{***}$ | $0.28$ |
| Suffix Push × Orthogonal shift | $-0.60^{***}$ | $-0.31^{***}$ | $0.41$ | $-0.06^{***}$ | $-0.18^{*}$ | $0.34$ |
| Constant | $-2.46^{***}$ | $-0.86^{***}$ | $-5.20^{***}$ | $-4.53^{***}$ | $-5.26^{***}$ | $-7.89^{***}$ |
| $N$ | 800,000 | 8,000 | 8,000 | 800,000 | 8,000 | 8,000 |
| Pseudo $R^2$ | 0.28 | 0.069 | 0.21 | 0.13 | 0.27 | 0.16 |

*Note:* Stars denote statistical significance levels. $^{*}$ $p < 0.05$, $^{**}$ $p < 0.01$, $^{***}$ $p < 0.001$. Recall that smaller $p$ values reflects stronger evidence for the hypothesis in question.

The interaction effects are mostly substantially smaller than the main effects—so one should be careful not to read too much into the specific sign patterns. Figures 10 to 13 visualize these interactions effects. From the figures we can conclude that for Qwen and Vicuna, the interaction effects are more relevant than for the Llama models.

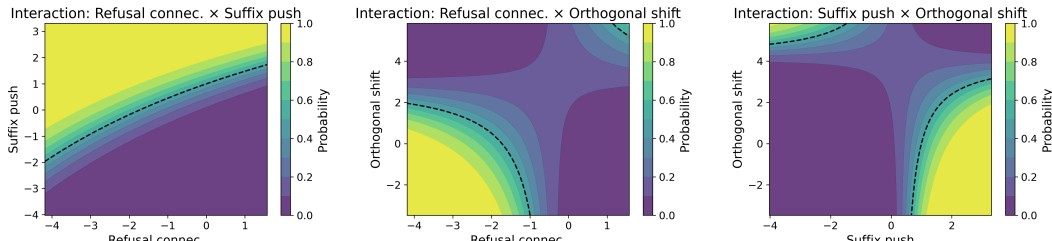

Figure 10: Visualization of the interaction effects in Table 7 for Qwen. "Proability" denotes the likelihood of a successful transfer given different levels of the main effects.

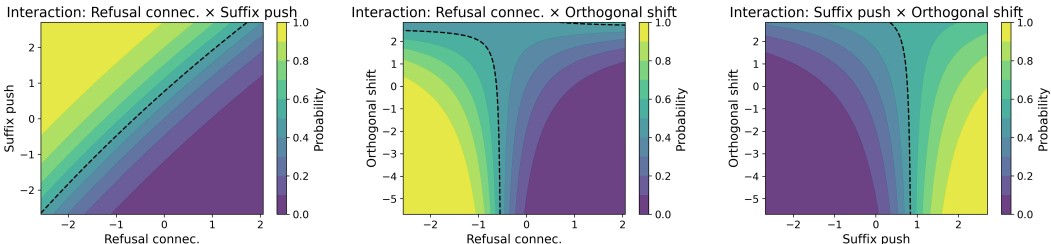

Figure 11: Visualization of the interaction effects in Table 7 for Vicuna. "Proability" denotes the likelihood of a successful transfer given different levels of the main effects.

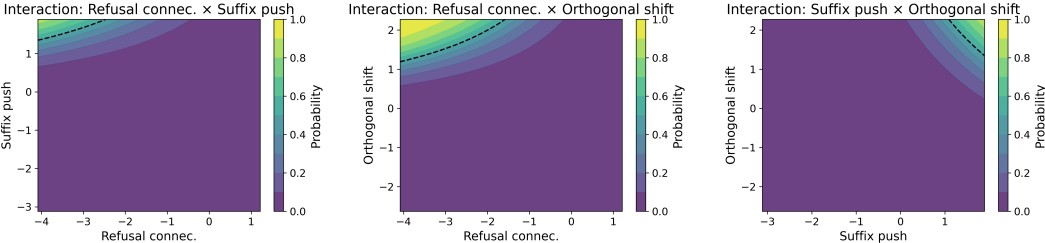

Figure 12: Visualization of the interaction effects in Table 7 for Llama 2. "Proability" denotes the likelihood of a successful transfer given different levels of the main effects.



Figure 13: Visualization of the interaction effects in Table 7 for Llama 3.2. "Proability" denotes the likelihood of a successful transfer given different levels of the main effects.

Table 8 displays the same logistic regression model with an added coefficient for semantic similarity. Semantic similarity is calculated as the similarity of embeddings between two prompts (as described in Definition 3). In this regression analysis, we use the semantic similarity based on model internal activations on the last instruction token at the layer where the refusal direction is extracted.

We observe that semantic similarity has a positive and highly statistically significant effect on transfer success (except for Llama 2), which means that if two prompts have high similarity in activations, their suffixes are more likely to successfully transfer. However, compared to the size of the coefficients for

suffix push and refusal connectivity, the influence is relatively small especially for Qwen and Vicuna, while comparably large in Llama 3.2. Again, the interaction effects are small in size compared to the main effects, except for Llama 2. However, given that for this model, the main effect for semantic similarity is not statistically significant, one should be cautious to interpret interaction effects based on this feature.

Table 8: Logistic regression coefficients (standardized) predicting transfer success (intra-model) including semantic similarity based on model internal embeddings.

| Variable | Qwen | Vicuna | Llama 2 | Llama 3.2 |
|---|---|---|---|---|
| Semantic sim. (model) | 0.06*** | 0.51*** | 0.26 | 0.42*** |
| Refusal connec. | −1.40*** | −1.45*** | −0.19 | −0.43*** |
| Suffix push | 2.39*** | 0.93*** | 1.68*** | 0.86*** |
| Orthogonal shift | 0.22*** | 0.32*** | 1.32*** | 0.49*** |
| Refusal connec. × Suffix push | −0.08** | −0.10** | −0.31* | −0.27*** |
| Refusal connec. × Orthogonal shift | 0.56*** | 0.49*** | −0.78** | 0.20*** |
| Suffix push × Orthogonal shift | −0.62*** | −0.30*** | 0.21 | −0.06*** |
| Semantic sim. (model) × Refusal connec. | −0.03*** | 0.38*** | 0.78*** | −0.04*** |
| Semantic sim. (model) × Suffix push | 0.36*** | −0.04 | −1.71*** | 0.00 |
| Semantic sim. (model) × Orthogonal shift | −0.14*** | 0.13** | 0.92*** | −0.08*** |
| Constant | −2.43*** | −0.95*** | −5.46*** | −4.61*** |
| $N$ | 800,000 | 8,000 | 8,000 | 800,000 |
| Pseudo $R^2$ | 0.288 | 0.117 | 0.227 | 0.140 |

*Note:* Stars denote statistical significance levels. * $p < 0.05$, ** $p < 0.01$, *** $p < 0.001$. Recall that smaller $p$ values reflects stronger evidence for the hypothesis in question.

**Prompt rephrasing instructions**   In Section 5.6 we use Vicuna to rephrase the harmful questions without changing their meaning. To generate diverse rephrases of jailbreak prompts, we use the following template prompt:

```
In the following you will get a prompt.  Please
generate {num_rephrases} rephrases of this prompt
that have the same meaning and will lead to a
similar answer.  You can add paraphrases that contain
setting the prompt into a hypothetical scenario, too,
like "hypothetically speaking, ...".  Each of the
{num_rephrases} rephrases should be unique.

Original prompt:  {prompt}

Please provide exactly {num_rephrases} rephrases, each
on a new line, numbered 1-{num_rephrases}:
```

In this template:

- {num_rephrases} is replaced with the desired number of rephrases to generate
- {prompt} is replaced with the original jailbreak prompt to be rephrased

This systematic approach ensures consistent generation of semantically equivalent variants while maintaining the adversarial intent of the original prompts.

