# OpenReview forum: "Towards Understanding the Transferability of Adversarial Suffixes in Large Language Models"
_ICLR.cc/2026/Conference — ICLR 2026 Conference Withdrawn Submission_

### Official Review · Reviewer_ssRD · 2025-10-25

**Soundness:** 3
**Presentation:** 3
**Contribution:** 1
**Rating:** 2
**Confidence:** 5

**Summary:**

This work investigates the transferability of adversarial jailbreak suffixes by analyzing their impact on a model's internal "refusal direction" vector. The study finds that transfer success is positively associated with suffix push and orthogonal shift. Conversely, transfer is negatively associated with "refusal connectivity," which is how much the original prompt aligns with the refusal direction. The authors also conclude that semantic similarity between prompts is only a weak predictor of transfer success.

**Strengths:**

This paper provides many interesting empirical analyses on suffix-based jailbreak attacks. For example, it identifies specific factors that strongly correlate with successful suffix-based transfer like suffix push and refusal connection. For another example, the analysis shows that the semantic similarity of a pair of prompts does not mean that a suffix originally generated for one prompt can more easily transfer to another one; this was found to be only a weak correlation.

**Weaknesses:**

1. The paper's key discovery about the refusal direction and orthogonal shift is essentially identical to what Rep Steering[1] already showed. Rep Steering uses hidden representations in jailbreak attacks to push attacks toward the acceptance direction. They also ran experiments proving that jailbreaks work because they shift harmful prompts to look more like harmless ones in the representation space, using their proposed method. They even tested their conclusion on more advanced methods like AutoDAN.

2. The experiments only test GCG-type suffixes. The paper needs to include tests with human-readable attack methods like AutoDAN.

3. The experiments only use two small models: Qwen2.5-3B and Llama3.2-1B. Why didn't the authors test on larger models with better safety features? This is a significant limitation.

[1] Towards Understanding Jailbreak Attacks in LLMs: A Representation Space Analysis

**Questions:**

1. How does your refusal direction finding differ from Rep Steering's existing work on representation shifts?

2. Why test only GCG-related suffixes and not human-readable attacks like AutoDAN?

3. Why test only on small models like Qwen2.5-3B and Llama3.2-1B instead of larger, safer variants?

---

### Official Review · Reviewer_J1KW · 2025-11-01

**Soundness:** 3
**Presentation:** 3
**Contribution:** 3
**Rating:** 6
**Confidence:** 4

**Summary:**

This paper investigates the mechanisms underlying the transferability of adversarial suffixes that can jailbreak LLMs when appended to harmful prompts. The authors conduct a large-scale statistical and interventional analysis and identify three geometric factors that correlate strongly with transfer success, refusal connectivity, suffix push, and orthogonal shift. Through experiments across several instruction-tuned models, the paper demonstrates that suffix push and orthogonal shift are consistently predictive of transfer success, while high refusal connectivity decreases transferability. Interventional experiments such as prompt rephrasing and modified GCG loss functions validate these insights and show how they can improve attack efficacy.

**Strengths:**

-	The paper presents a systematic and mechanistic study of adversarial suffix transferability in LLMs. This paper formalizes and quantifies it through interpretable geometric features linked to internal representations.
-	The work is methodologically sound and well-controlled. Definitions of geometric quantities are clear and well-grounded in prior interpretability research on refusal directions.
-	The findings provide a conceptual and practical framework for understanding why some jailbreaks generalize across prompts and models.
-	The paper is clearly written and logically structured.

**Weaknesses:**

-	Experiments are limited to small-sized open models and a single dataset. Whether the conclusions generalize to larger frontier models or to more diverse jailbreak scenarios, such as multi-turn or multilingual settings, is unknown.
-	Inter-model asymmetries (e.g., Llama to Qwen vs. Qwen to Llama) are observed but not fully analyzed. A more detailed discussion of how alignment strength or architectural factors influence these asymmetries would strengthen the argument.
-	Missing discussion with existing work on adversarial suffix transferability [1,2].

[1] Advancing adversarial suffix transfer learning on aligned large language models. EMNLP 2024.

[2] Unnatural Languages Are Not Bugs but Features for LLMs. ICML 2025.

**Questions:**

-	Do larger or more strongly aligned models exhibit similar geometric relationships, or does stronger alignment reduce transferability via altered refusal representations? Any insight on how the way of model training/alignment contribute to these factors for transferability?
-	Have the authors tested whether the same correlations hold for other harmful prompt distributions or multi-turn jailbreaks?

---

### Official Review · Reviewer_zopi · 2025-11-02

**Soundness:** 2
**Presentation:** 3
**Contribution:** 2
**Rating:** 2
**Confidence:** 5

**Summary:**

```
This paper empirically analyzes how different factors could affect the transferability (across prompts or models) of jailbreak suffixes synthesized via ONLY the GCG attack. Main findings include: (1) jailbreak robustness of LLMs (or "prompts") might have connection with the LLM refusal direction, (2) semantically similar prompts do not lead to similar jailbreak suffixes. These findings may help the ML safety community to better understand the mechanism behind suffix jailbreak attacks.
```

**Strengths:**

```
This paper provides many interesting empirical analyses on suffix-based jailbreak attacks that I appreciate. For example, jailbreak suffixes can easily transfer within a single model but are difficult to transfer across different models. For another example, the semantic similarity of a pair of prompts does not mean that a suffix originally generated for one prompt can more easily transfer to another one.
```

**Weaknesses:**

```
1. A major weakness of this paper is that the authors only leverage a single jailbreak attack named GCG to conduct their empirical investigation, which is not enough. I believe that verifying the obtained findings on various jailbreak attacks beyond GCG attack is very important.

2. Furthermore, one should also notice that GCG can only reproduce gibberish suffix texts to conduct attacks. So it remains unknown whether conclusions obtained from GCG would also hold for human-readable jailbreak suffixes. I suggest the authors adopt additional suffix jailbreak attacks that can produce **semantic** jailbreak prompts in their experiments such as BEAST [r1], Zhu's AutoDAN [r2], AmpleGCG [r3], AdvPrompter [r4], etc.

3. In Definition 4 "Refusal connectivity", the authors propose to use the cosine similarity between the activation vector $a_i^{base}$ and the refusal direction $v_{refusal}$ to analyze/measure the jailbreak robustness of LLMs (i.e., Figure 4 in Section 5.3). I cannot see the motivation behind this. The refusal direction is defined as the difference between the (averaged) normal activation vector and the (averaged) refusal activation vector, why would one want to analyze the similarity between the activation vector and the "activation vector difference"? After all, how can a "(prompt's) activation feature" be compared with a "direction" in the activation space?


**References**

[r1] Sadasivan et al. "Fast Adversarial Attacks on Language Models In One GPU Minute". ICML 2024.

[r2] Zhu et al. "AutoDAN: Interpretable Gradient-Based Adversarial Attacks on Large Language Models". COLM 2024.

[r3] Liao et al. "AmpleGCG: Learning a Universal and Transferable Generative Model of Adversarial Suffixes for Jailbreaking Both Open and Closed LLMs". COLM 2024.

[r4] Paulus et al. "AdvPrompter: Fast Adaptive Adversarial Prompting for LLMs". ICML 2025.
```

**Questions:**

```
See **Weaknesses**.
```

---

### Official Review · Reviewer_GpgR · 2025-11-03

**Soundness:** 3
**Presentation:** 3
**Contribution:** 2
**Rating:** 2
**Confidence:** 4

**Summary:**

This paper investigates the mechanism underlying the transferability of adversarial suffixes in large language models (LLMs). The authors conduct a statistical and interventional analysis to understand why certain adversarial suffixes—optimized on one prompt or model—can successfully jailbreak others.
The study introduces three geometric factors derived from the models’ activation space—refusal connectivity, suffix push, and orthogonal shift—and quantifies their influence on transfer success through large-scale experiments.

**Strengths:**

The paper features a comprehensive empirical setup, covering multiple model families and including both intra-model and inter-model transfer analyses, which enhances the generalizability of the conclusions.
Its findings have direct implications for understanding and improving the robustness (or conversely, the design) of jailbreak attacks against LLMs.

**Weaknesses:**

A key limitation lies in the lack of deeper mechanistic explanation. While the correlations between geometric features and transfer success are clearly demonstrated, the causal reasoning behind why these specific directions influence model behavior remains insufficiently explored. In particular, the paper does not explain why a larger “suffix push” geometrically leads to semantic compliance or behavioral change in the model.

**Questions:**

1. On semantic similarity experiments: Since the GCG-generated suffixes are largely meaningless, it seems plausible that they contribute minimally to the embedding similarity between original and adversarial prompts. Could the authors visualize or report the embedding distribution of base versus adversarial prompts to validate this assumption?
2. On Figure 5: The observation that adding the least successful suffixes only marginally reduces refusal alignment, while the most successful suffixes cause a significant drop, appears somewhat counterintuitive. Could the authors provide further explanation —e.g., examining dependence on layer depth or suffix length—to clarify the underlying cause?

---

### Note · Authors · 2025-11-25

I have read and agree with the venue's withdrawal policy on behalf of myself and my co-authors.